# Highlighting Recent Crystalline Engineering Aspects of Luminescent Coordination Polymers Based on F-Elements and Ditopic Aliphatic Ligands

**DOI:** 10.3390/molecules27123830

**Published:** 2022-06-14

**Authors:** Richard F. D’Vries, Germán E. Gomez, Javier Ellena

**Affiliations:** 1Facultad de Ciencias Básicas, Universidad Santiago de Cali, Calle 5 # 62-00, Cali 760035, Colombia; 2Instituto de Investigaciones en Tecnología Química (INTEQUI), Área de Química General e Inorgánica, Facultad de Química, Bioquímica y Farmacia, Chacabuco y Pedernera, Universidad Nacional de San Luis, Almirante Brown, 1455, San Luis 5700, Argentina; germangomez1986@gmail.com; 3São Carlos Institute of Physics, University of São Paulo, São Carlos CEP 13.566-590, SP, Brazil; javiere@ifsc.usp.br

**Keywords:** coordination polymers, f-elements, luminescence

## Abstract

Three principal factors may influence the final structure of coordination polymers (CPs): (i) the nature of the ligand, (ii) the type and coordination number of the metal center, and (iii) the reaction conditions. Further, flexible carboxylate aliphatic ligands have been widely employed as building blocks for designing and synthesizing CPs, resulting in a diverse array of materials with exciting architectures, porosities, dimensionalities, and topologies as well as an increasing number of properties and applications. These ligands show different structural features, such as torsion angles, carbon backbone number, and coordination modes, which affect the desired products and so enable the generation of polymorphs or crystalline phases. Additionally, due to their large coordination numbers, using 4*f* and 5*f* metals as coordination centers combined with aliphatic ligands increases the possibility of obtaining different crystal phases. Additionally, by varying the synthetic conditions, we may control the production of a specific solid phase by understanding the thermodynamic and kinetic factors that influence the self-assembly process. This revision highlights the relationship between the structural variety of CPs based on flexible carboxylate aliphatic ligands and f-elements (lanthanide and actinides) and their outstanding luminescent properties such as solid-state emissions, sensing, and photocatalysis. In this sense, we present a structural analysis of the CPs reported with the oxalate ligand, as the one rigid ligand of the family, and other flexible dicarboxylate linkers with –CH_2_– spacers. Additionally, the nature of the luminescence properties of the 4*f* or 5*f*-CPs is analyzed, and finally, we present a novel set of CPs using a glutarate-derived ligand and samarium, with the formula [2,2′-bipyH][Sm(HFG)_2_ (2,2′-bipy) (H_2_O)_2_]•(2,2′-bipy) (**α-Sm**) and [2,2′-bipyH][Sm(HFG)_2_ (2,2′-bipy) (H_2_O)_2_] (**β-Sm**).

## 1. Introduction

Coordination polymers are composed of a rational combination of metallic centers (connectors) and organic ligands, resulting in extended 1D, 2D, or 3D structures [1,2]. Many classifications have been assigned to these materials, depending on their structure, dimensionality, porosity, catalytic capacity, etc. [1,3,4]. However, in this work, we will focus on the nature of the aliphatic ligand in combination with 4*f* and 5*f* elements to yield a plethora of crystalline coordination polymers (CPs). These materials have been extensivelt studied in recent decades due to their multifunctional properties, which are intimately related to their structural features [5,6,7]. Properties such as luminescence, catalysis, sensing, gas storage, and drug delivery were thoroughly investigated, becoming mature areas [8,9,10,11,12,13,14,15,16,17,18,19,20,21]. 

Moreover, the use of metallic connectors from 4*f* and 5*f* metals allows the development of materials with unique optical properties derived from their pure color emission, fine line f-transitions, and variable lifetime values depending on the desirable applications [12,15,22,23]. Additionally, the high coordination numbers and the oxophilic nature of these metallic centers [24,25] allow us to combine these ions with flexible carboxylate aliphatic ligands and build a “toolbox” to construct families of novel crystalline structures [26]. This work discusses some outstanding examples in structural variety of CPs using ditopic flexible ligands [−OOC-(CH_2_)n-COO−] and the role of the oxalate ligand as the first member of the dicarboxylate family, focusing on their luminescent properties, as well as applications such as dye photocatalysis and chemical and thermal sensing in *Ln*-CPs or *An*-CPs (*Ln* = lanthanides and *An* = actinides). Additionally, we present two novel crystalline phases of CPs obtained from the combination of hexafluorglutaric acid (H_2_HFG) and samarium ions.

## 2. Discussion

### 2.1. Coordination Polymers Based on Oxalate Linker

In this highlight, we will analyze the entire family of aliphatic dicarboxylate ligands, from oxalate to dodecanedioate, used as ligands in the formation of CPs. The analysis on Cambridge Crystallographic Data Centre (CCDC) during the last two decades shows a small number of entries with a decreasing number of reported structures as the length of the carbon backbone increases (Figure 1). This trend is attributed to a higher degree of freedom and flexibility when chain length increases, which restricts or decreases the chance of crystallization.

In this sense, it is important to consider the nature of the sp^3^ covalent bond along the aliphatic chains, which has the ability to rotate along the C–C bond. This degree of freedom in the simple bond enables the formation of multiple conformers. The amount of conformers for a particular aliphatic linker is directly related to the carbon number and carboxylate coordination modes. The use of ditopic aliphatic ligands in the synthesis of CPs is limited by the stability of the conformer, the size of the ligand and the architecture of the polymer. Additionally, it is important to note the high number of reports with short dicarboxylates (carbon numbers between 2 and 5, Figure 1) forming *Ln*-CPs or *An*-CPs (*Ln* = lanthanides and *An* = actinides) [27,28,29,30]. The oxalate anion is not a flexible ligand and contributes with rigidity and stability in the resulting CP. In fact, this molecule has been found in biologic systems [31], in materials such as electronic and luminescent devices [32,33], drugs [34], and minerals [35,36]. Additionally, the use of this ligand in combination with Ln and An metals as a toolbox to obtain CPs dates back to the 1980s, where Kahwa et al. [37] reported a family of isostructural 1D structures with the general formula K_3_[Ln(ox)_3_(OH_2_)]·2H_2_O, (Ln = Nd, Sm, Eu, Gd, Tb). Additionally, Alexander et al. [38] reported a 2D terbium oxalate with lanthanide-centered green luminescence at 543 nm by exciting the sample at 369 nm. The observed signals correspond to the most intense forbidden *f–f* transition of terbium ions exhibiting an optical performance without ligand sensitization, comparable to the commercial green phosphors [38]. 

The general use of this ligand is extended to mixtures of two or more ligands using oxalate and auxiliary ligands. The combination of two types of linkers is extensively used since it allows the presence of a structural ligand and a functional one allowing the formation of CPs with improved properties such as luminescence or catalysis. Indeed, approximately 69% of the CCDC entries found in this work (Figure 1) present a combination of ligands. One example of this trend is the work by Xu et al. [39], where they used a combination of aromatic ligands as 4-(4-carboxyphenoxy)-isophthalic acid (cphtH_3_) and 1,10-phenanthroline (phen) as an ancillary to form a highly stable luminescent 3D CPs [39]. These compounds produce green (Tb), white (Sm), blue (Dy) and red (Eu) emissions. Additionally, based on the excellent luminescence of the *Eu*-MOF produced, the compound was tested for quercetin and Fe^3+^ ion sensing based on quenching processes [39].

Moreover, studies of *An*-CPs based on uranyl or thorium ions have been explored in the last decade, mostly due to their potential applications as light emitters for sensing and photocatalysis, and the variety of novel achieved architectures [22,40,41,42,43,44,45,46,47,48,49]. Furthermore, the rational combination of O-donor with the uranyl cation [UO_2_]^2+^ coordination modes has led to the formation of an important number of new organic–inorganic connectivity with different dimensionalities and nuclearities of uranyl-centered building units [50,51,52,53,54]. These assemblies are commonly obtained by employing O-donor-chelating agents such as polycarboxylic acids. In spite of that, the formation of An-CPs is scarce and just 5.9% of all the structures reported in the CCDC search present the oxalate anion (Figure 2). The formation of new structures with these components could open a wide research area of functional compounds derived from the nuclear activity [55,56].

### 2.2. Flexible Linkers with –CH_2_– Spacers

The torsion angle of the most used ligands in the synthesis of CP is shown in Figure 3, where the oxalate ligand is a planar ligand, whereas malonate and succinate show a wide range of O–C–C–O torsion angle values. Thus, the degree of freedom around the sp^3^ carbon as well as the coordination modes enables the formation of multiple phases with a slight change in the reactions conditions. Cañadillas-Delgado et al. [58], Chrysomallidou et al. [59], and Delgado et al. [29] reported that malonate compounds show different dimensionalities, coordination modes, and carbonyl–carbonyl torsion angles according to the synthetic methodology. These findings describe the synthesis of compounds with a 2D and 3D topology by solvothermal reactions, slow diffusion of the reagents in metasilacate gel and slow evaporation of the solvent. From a crystal engineering point of view, it is possible to modify the structural features using different synthetic methodologies, reaction conditions and use of ancillary ligands [60]. Within the most used synthetic techniques are (i) hydro or solvothermal synthesis, (ii) slow solvent evaporation, (iii) gel diffusion, (iv) mechanochemical, and (v) microwave-assisted synthesis [61,62,63,64,65,66,67]. It has been observed that the use of methodologies that involve the application of energy favors the formation of structures that are more compact, with more complex coordination modes and high dimensionalities [68]. In the case of methodologies involving low energy, they generate low-dimensionality structures with simpler coordination modes. On other hand, the use of *guest molecules acting as templates* refers to the presence of organic molecules or solvent molecules giving space for the formation of cavities into the coordination polymer [69]. Guest molecules can be small and isolated species included in the CP structure and non-coordinated to the metal that allow the formation of cavities [67,70,71,72].

In our group, we have carried out studies involving the formation of four different phases of *Ln*-succinate compounds by solvothermal reaction conditions [73], using the *template effect* or *guest molecule acting as a template* of aromatic solvents as toluene, and aromatic organic molecules as 5-sulfosalicylate (5-SSA^3−^). These guest molecules are directly involved in the formation of CPs with large pores or cavities, unlike using conventional protic or organic solvents. Additionally, interesting reviews dedicated to the formation of CPs based on succinate ligands with 2D and 3D structures involving different topologies and applications, show the use of these type of ligands in the design of multifunctional CPs [74]. The use of aliphatic linkers such as 2-methylsuccinates and 2-phenysuccinates can efficiently separate the lanthanide ions in order to avoid concentration quenching and giving rise lanthanide-centered emissions upon direct excitation into the 4*f* levels. The fine lanthanide emissions were enough to use the materials as solid-state emitters, thermal sensors and chemical sensors for small molecules [42,75]. Additionally, it is important to highlight the variety of coordination modes found reports employing flexible ditopic ligands [55,76,77], as shown in Figure 4.

### 2.3. The Luminescent Properties of 4f-5f Compounds 

According to the vast literature on luminescent materials, their properties can be explored by the following studies [78]: (1) excitation and emission spectra; (2) quantum yield (QYs) determinations, and (3) experimental or observed lifetime (τ) of emission. Additionally, the attenuation of luminescence experienced by certain materials is known as *quenching of luminescence* and can be derived from structural features as well as from external parameters. The contributions of non-radiative pathways mainly include electron transfer quenching, back-energy transfer to the sensitizer as well as quenching by organic vibrations from the linkers (C-H, O-H, N-H) [79]. Among lanthanide ions, Sm^3+^, Eu^3+^, Gd^3+^, Tb^3+^ and Dy^3+^ are preferred ions for optical device implementation and optoelectronics due to their intense, long-lived and fine emissions into the visible region [12,32,80]. Additionally, Er^3+^, Ho^3+^, Yb^3+^ and Tm^3+^ ions are suitable ions for up-conversion emissions into visible and near-infrared regions [81,82].

The complete review by Bernini and colleagues highlights the contributions of diverse lanthanide-succinate-derived structures with applications in solid-state lighting, sensing, and catalysis [74]. In this report, the authors mention the impact of aryl and alkyl substitute succinate ligands on the final dimension of the crystalline structure and also on the final property. On the other hand, the improved performance can be assessed by a correct selection of building blocks that allows a correct energy gap between the lanthanide ions and the exited states (singlet or triplet) from the linkers. 

In general, materials constructed by aryl-derivate linkers or by incorporating auxiliary aromatic ligands (i.e., 1,10-phenanthroline, 2,2′-bipyridine, etc.) show better emission efficiency than the solely ditopic-based materials [53]. One example is phthalate ligand sensitization to improve Eu/Tb luminescence and metal-to-metal energy transfer in mixed [Ln(adipate)_0.5_(phth)(H_2_O)_2_] compounds, being used as thermometers in the 303–423 K range [83].

From 2018, we can mention remarkable contributions employing lanthanide-succinate compounds by Professor Narda’s group. They report the synthesis of a bidimensional mixed structure Tb^3+^@Y-succ-sal (succ = succinate, sal = salycilate) with the particularity of producing reactive oxygen species upon UV excitation in aqueous suspensions, allowing the photodynamic inactivation of *Candida albicans* culture by intersystem crossing mechanisms [84]. In 2022, the same group reported the use of doped Yb^3+^/Er^3+^/Gd^3+^@Y-succ-sal systems as sacrificial materials to obtain Yb^3+^/Er^3+^/Gd^3+^@Y_2_O_3_ and finally deposit them onto glass substrates for thin film implementation (Figure 5) [85]. The red emissions derived from up-conversion processes by exciting into the near-infrared region yielded τ values ranging from 144 to 300 μs. 

Further, uranyl emission is originated from a ligand to metal charge transfer (LMCT) by exciting an electron from non-bonding 5*f*_δ_, 5*f*_φ_ uranyl orbitals to uranyl–oxygen bonding orbitals (σ_u_, σ_g_, π_u_, π_g_) [86], which is further coupled to “yl” vibrational (S_11_ → S_01_ and S_10_ → S_0ν_ [ν = 0–4]) states of the U=O axial bond [87]. Its phosphorescence is often characterized by green emission, which manifests as four to six vibronically coupled peaks (up to 12) in the 400–650 nm range.

As shown in lanthanide-flexible compounds, uranyl versions can be explored in interesting applications such as photocatalysis and sensing. This is achievable due to its high water stability, repeatability, and bright emission into the visible region, showing a bright future for uranyl-CPs.

In this case, we can highlight the **UNSL-1** (Universidad Nacional de San Luis) compound, [(UO_2_)_2_(phen)(succ)_0.5_(OH) (μ_3_-O)(H_2_O)]∙H_2_O, which corresponds to a 1D coordination polymer formed by tetramer units of uranyl ions, decorated by coordinated 1,10-phenanthroline molecules and connected by succinate ligands into the [–1 0 1] direction [22]. The phen plays the role of a suitable antenna molecule to store UV energy and then transfer it to the uranyl ions, yielding a bright green emission into the visible region (see Figure 6). For these optical features, **UNSL-1** material was employed as a photocatalyst for methylene blue degradation upon sunlight excitation. Additionally, the material was used as a sensor toward metallic ions in aqueous media, exhibiting sensitivity under Fe^2+^ ions.

### 2.4. Sm-Hexafluoroglutarato CPs

Glutaric acid is a long, flexible carboxylic ligand with –CH_2_– spacers (polymethylene groups) with a size chain of approximately 7 Å. Similarly, the glutarate presents a wide range of C–C–C–O torsion angles as well as different configurations around the methylene scaffold [88,89,90]. According to Kumar et al. [30], it is interesting to differentiate between two types of compounds: (i) those without ancillary co-ligands and (ii) those with ancillary co-ligand [30]. Further, there is one report of complexes formed with H_2_HFG ligand [91], with Eu^3+^ and Tb^3+^ ions giving rise to dimeric assemblies with the formula [Ln_2_(HFG)_2_(phen)4(H_2_O)_6_]. · HFG · 2H_2_O (where Ln = Eu or Tb). In the mentioned work, the phen molecule locks the coordination positions in the metal ions, limiting the dimensionality of the final product. 

Here, we present a novel set of CPs using a glutarate-derived ligand and samarium, with the formulae [2,2′-bipyH][Sm(HFG)_2_ (2,2′-bipy) (H_2_O)_2_]•(2,2′-bipy) (**α-Sm**) and [2,2′-bipyH][Sm(HFG)_2_ (2,2′-bipy) (H_2_O)_2_] (**β-Sm**). The ORTEP diagrams of the asymmetric units of both compounds are shown in Figure 7, and the crystallographic and refinement data are shown in Table 1.

In both cases, **α-Sm** and **β-Sm** are formed by one crystallographically independent nine-coordinated Sm(III) center surrounded by two nitrogen atoms from a 2,2′-bipy, and seven oxygen atoms belonging to two HFG ligands and two water molecules to form a distorted trigonal prism, square-face tricapped polyhedron. In the first case, it is possible to observe one free 2,2′-bipy and a half protonated 2,2′-bipy (2,2′-bipyH) in the asymmetric unit, while just one protonated (2,2′-bipyH) molecule is presented in **β-Sm**. However, in both cases, the HFG ligand acts as a bridge between metallic centers through the μ_2_κ^3^ and μ_2_κ^2^ coordination modes, giving raise to 1D CPs. In **α-Sm**, chains grow along the [0 0 1] direction, whereas chains in **β-Sm** grow along the [1 1 1] direction (Figure 8). The presence of 2,2′-bipy and 2,2′-bipyH is observed in the inter-chain space, where they play role of counter ions as well as in the stabilization of the crystal packing. 

As was previously mentioned, the synthesis conditions as well as the flexibility of the ligand and coordination modes determine the presence of different crystalline phases. In this case, the reaction conditions allow the formation of two crystalline phases, where there are clearly observed structural differences around the HFG ligand. Both structures have two HFG ligands coordinated to the metal center, but the carboxylate torsion angle with respect to the C–C scaffold significantly changes, as it is presented in Figure 9.

## 3. Materials and Methods

A Cambridge Structural Database (CSD) search was performed in CSD 2021.2 (November 2021) using Conquest software (version 2021.20). To compute *Ln* and *An* complexes with ditopic ligands, we used the formula [M(ditopic)] as a query in the general search.

### 3.1. Synthesis of [2,2′-bipyH][Sm(HFG)_2_ (2,2′-bipy) (H_2_O)_2_]•(2,2-bipy) and [2,2′-bipyH][Sm(HFG)_2_ (2,2′-bipy) (H_2_O)_2_]

Both compounds were synthesized by mixing Sm(NO_3_)_3_.6H_2_O, H_2_HFG and 2,2′-bipyridine in a 0.1:0.15:0.2 millimolar ratio in 4 mL of water. After that, a white suspension was achieved and one drop of concentrated nitric acid (65%) was added in order to obtain a transparent solution. Then, the mixture was left to stand at room temperature for slow solvent evaporation. After three months, the crystalline product was washed with distilled water and dried at room temperature (yield: 45 mg).

### 3.2. Single-Crystal X-ray Diffraction (SCXRD) for Structure Determination

SCXRD data for **α-Sm** were collected at room temperature (293(2) K) using MoK_α_ radiation (0.71073 Å) monochromated by graphite on a Rigaku XTALAB-MINI diffractometer. The unit cell determination as well as the final cell parameters were obtained on all reflections using *CrysAlisPro* software [92]. Data collection strategy, integration and scaling were performed using *CrysAlisPro* software [92]. **Β-Sm** was collected at room temperature (293(2) K) using MoK_α_ radiation (0.71073 Å) monochromated by graphite on an Enraf–Nonius Kappa-CCD diffractometer. The initial cell refinements were performed using the software Collect [93] and Scalepack [94], and the final cell parameters were obtained on all reflections. Data reduction was carried out using the software Denzo-SMN and Scalepack [94].

The structures were solved and refined with SHELXT [95], and SHELXL [96], software, respectively, including in Olex2 [97]. In all cases, non-hydrogen atoms were clearly resolved and full-matrix least-squares refinement with anisotropic thermal parameters was performed. In addition, hydrogen atoms were stereochemically positioned and refined using the riding model in all cases [98]. The Mercury [99] program was used for the preparation of artwork. The structures were deposited in the CCDC database under the codes 2159499-2159500. Copies of the data can be obtained, free of charge, via www.ccdc.cam.ac.uk (accessed on 16 May 2022).

## 4. Conclusions

Although considerable research has been conducted on the design of new CPs, the use of flexible ditopic aliphatic ligands as structural support or even as a sensitizer for luminescence remains a viable strategy to developing a new generation of optically efficient *Ln*-CP- and *An*-CP-based materials. Combining aliphatic and aromatic auxiliary ligands or space holders seems to be a promising route for developing novel materials with luminescent properties. Additionally, the synthesis conditions as well as the diverse methodology approaches determine the variations in the promotion of ligand motion, which may result in the formation of different crystal structures and dimensionalities. One example of the use of a flexible ditopic ligand is shown in this work, where two CPs obtained at room temperature employ a hexafluoroglutarate ligand and samarium, [2,2′-bipyH][Sm(HFG)_2_ (2,2′-bipy) (H_2_O)_2_]•(2,2′-bipy) (**α-Sm**) and [2,2′-bipyH][Sm(HFG)_2_ (2,2′-bipy) (H_2_O)_2_] (**β-Sm**), and are reported for the first time herein. Although the chemistry of flexible ligand-based CPs is in constant growth, few studies on their incorporation into composites have been reported. As such, the future of *Ln*- and *An*-CPs is bright in terms of mesmerizing crystalline structures and exciting applications over the next decades.

## Figures and Tables

**Figure 1 molecules-27-03830-f001:**
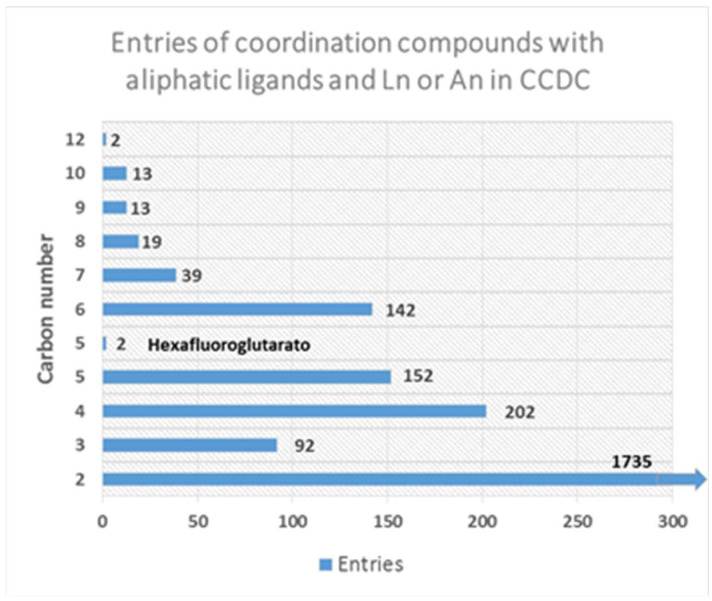
Entries distribution in the CCDC for compounds with aliphatic ditopic ligands and Ln or An metals.

**Figure 2 molecules-27-03830-f002:**
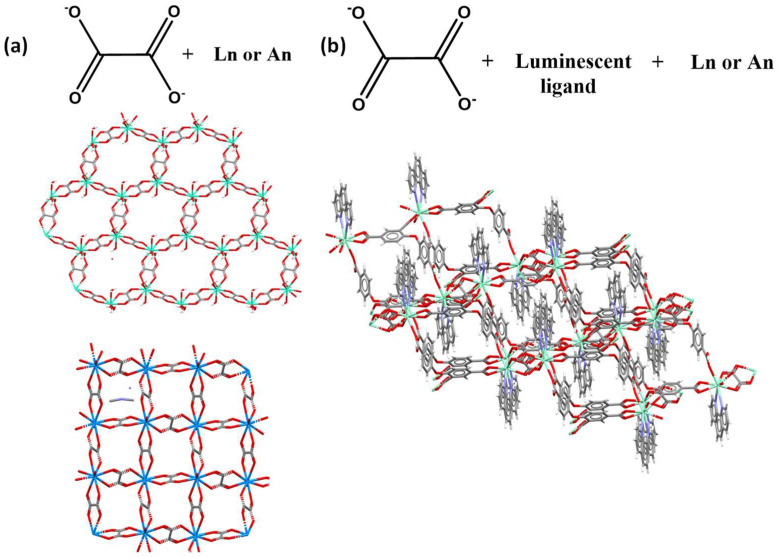
(**a**) Terbium [38] And uranium [57] 2D oxalate CPs and (**b**) lanthanide−oxalate−phen-cphtH_3_ 3D CP [39].

**Figure 3 molecules-27-03830-f003:**
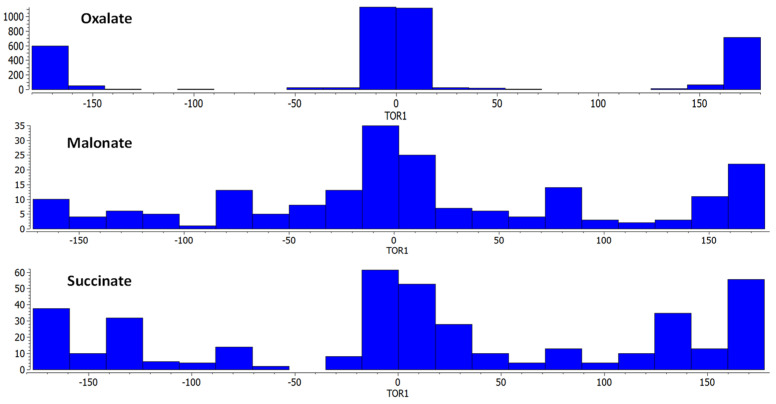
Torsion angle values for the entries in the CCDC for oxalate, malonate and succinate ligands.

**Figure 4 molecules-27-03830-f004:**
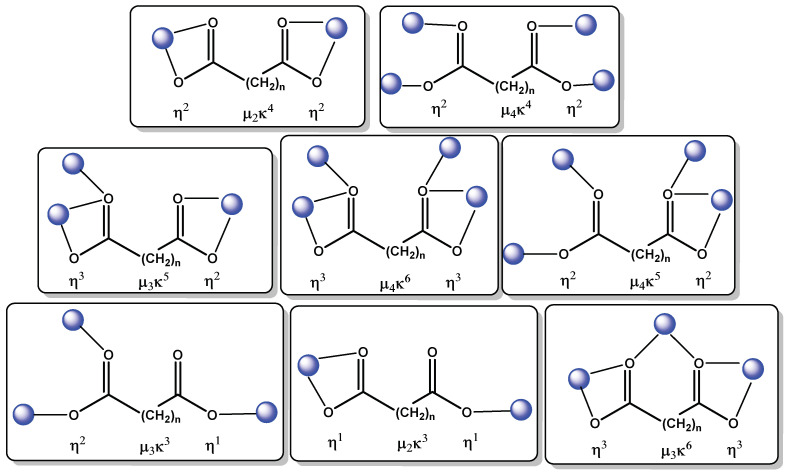
Typical coordination modes observed in ditopic aliphatic ligands.

**Figure 5 molecules-27-03830-f005:**
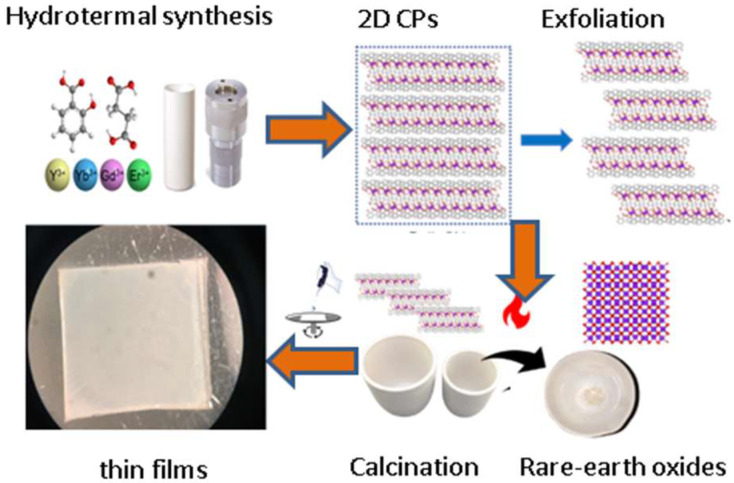
Manufacture of thin films based on bidimensional Y-succ-sal compounds: solvothermal synthesis of layered compounds, exfoliation, calcination to obtain lanthanide-doped Y_2_O_3_ systems and deposition by spin coating. Adapted from Ref. [85].

**Figure 6 molecules-27-03830-f006:**
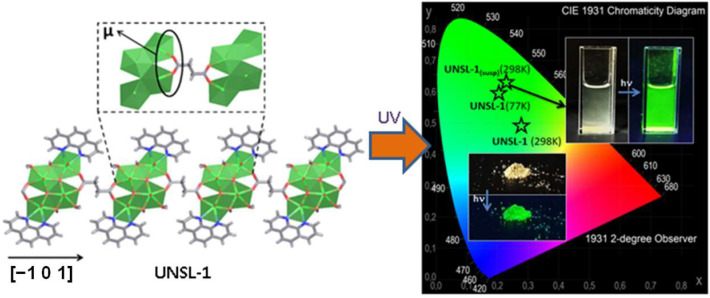
View of the infinite chains along the [−1, 0, 1] direction. CIE x, y chromaticity of **UNSL-1** compound in the suspension and solid states (77 and 298 K), Adapted from Ref. [18].

**Figure 7 molecules-27-03830-f007:**
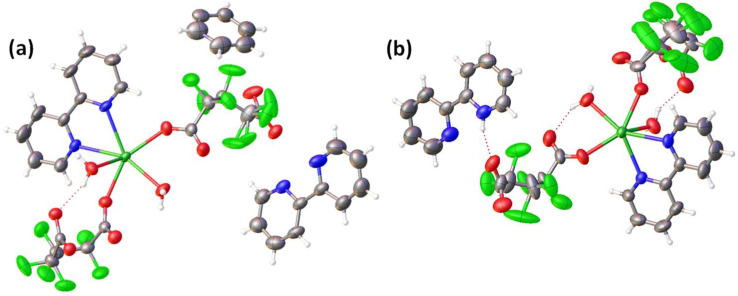
ORTEP-type diagrams with 50% of ellipsoid probability of the compounds (**a**) **α-Sm** and (**b**) **β-Sm**.

**Figure 8 molecules-27-03830-f008:**
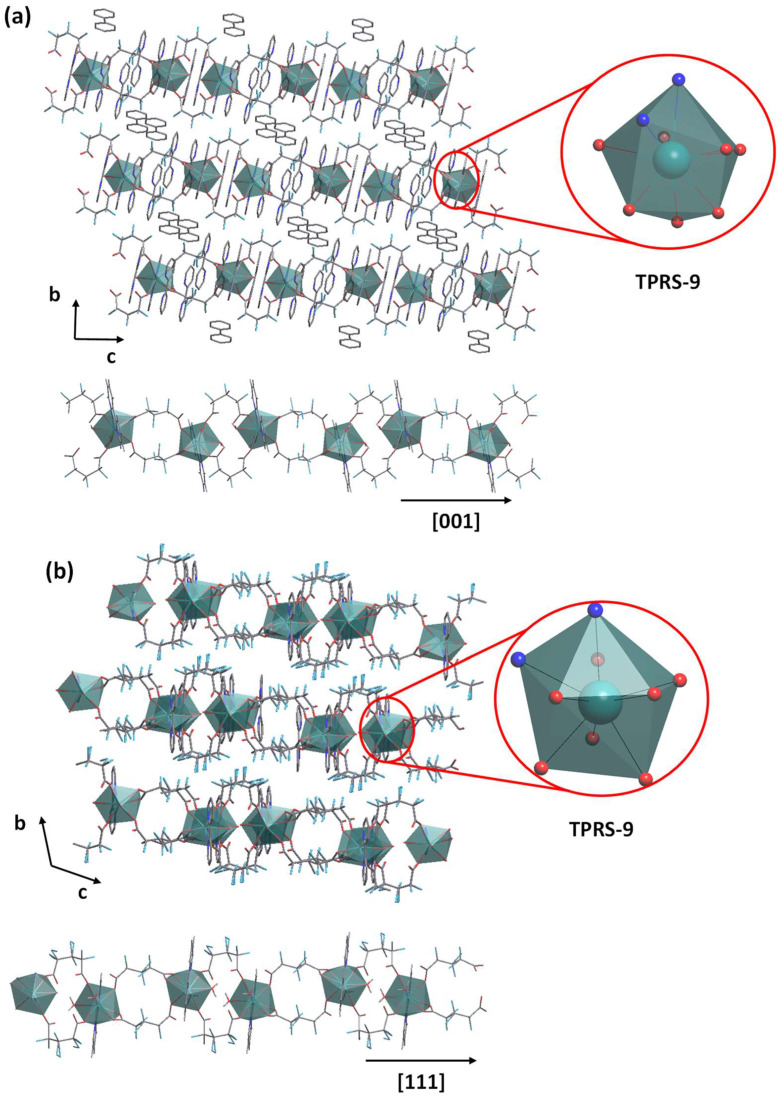
Crystal packing: 1D chains and coordination polyhedron for the compounds (**a**) **α-Sm** and (**b**) **β-Sm**.

**Figure 9 molecules-27-03830-f009:**
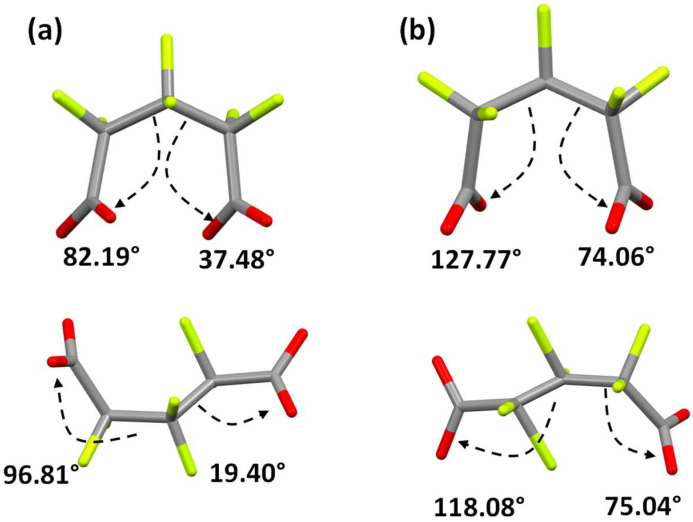
Torsion angles in the HFG ligands observed in the asymmetric unit for the compounds (**a**) **α-Sm** and (**b**) **β-Sm**.

**Table 1 molecules-27-03830-t001:** Crystallographic data and refinement parameters for **α-Sm** and **β-Sm** compounds.

	α-Sm	β-Sm
**Empirical formula**	C_35_H_24_F_12_N_5_O_10_Sm	C_30_H_21_F_12_N_4_O_10_Sm
**Formula weight (g/mol)**	1052.94	975.86
**Crystal system**	triclinic	triclinic
**Space group**	Pī	Pī
** *a* ** **/Å**	10.5754(4)	10.7751(10)
** *b* ** **/Å**	12.8624(5)	13.3601(15)
** *c* ** **/Å**	15.5452(7)	14.5413(17)
**α(°)**	70.371(4)	105.535(5)
**β(°)**	77.786(3)	98.863(5)
**γ(°)**	76.070(3)	114.104(5)
**Volume/Å^3^**	1913.23(13)	1756.4(3)
**Z**	2	2
**ρ_calc_ mg/mm^3^**	1.828	1.845
**μ/mm^−1^**	1.658	1.797
**F(000)**	1038	958
**2θ range for data collection/°**	5.04 to 69.18	6.31 to 51
**Reflections collected**	49,281	11,855
**Independent reflections**	15,269	6533
**Data/restraints/parameters**	15,269/12/570	6533/676/561
**Goodness of fit on F^2^**	1.161	0.969
**Final R index [I>2σ(I)]**	0.0550	0.0522
**Largest diff. Peak/hole/e.Å^−3^**	1.75/−1.66	1.13/−1.00

## Data Availability

Not applicable.

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
