# Peer review of "Highlighting Recent Crystalline Engineering Aspects of Luminescent Coordination Polymers Based on F-Elements and Ditopic Aliphatic Ligands"

_molecules, 2022, doi:10.3390/molecules27123830_

Round 1

Reviewer 1 Report

In this review, the authors mainly summarized the progress in crystalline engineering aspects for f-elements- and ditopic-aliphatic-ligand based luminescent coordination polymers. Some important aspects, such as coordination polymers based on oxalate linker, flexible linkers with –Ch2-spacers, luminescent properties of f-element contained compounds were reviewed and discussed. Moreover, the perspective was also commented. After carefully reading full text, I feel that this manuscript is acceptable for publication after proper revisions.

1/ In the abstract, the main points of this review should be included, and some general comments on the progress in this topic should also be added.

2/ A paragraph which presents the main content of this review should be added as the last paragraph of the INTRODUCTION section.

3/ The title of section 2 should be revised. Moreover, it is better to number the sub-sections in section 2.

4/ Section 3 is significant. However, there are a few words for this section. The authors should read more published papers and add some new content for this section.

5/ It is not satisfactory that only 59 references were cited. Some more references are asked for citations. Additionally, the references should be carefully formatted based on the journal criteria.

6/ In general, English seems good to me. However, there are still some bad sentences and grammar errors. Therefore, the authors should carefully polish their English.

Author Response

Dear  Felicia Yao

Assistant Editor

Molecules

Dear Felicia

In the following document I send to you the answers and modifications for the submitted Viewpoint to Molecules (1774737).

According to the strong suggestion of the referees, we put major efforts to considerably improve the manuscript to be accepted in Molecules. In this sense, we have carefully considered all comments and all raised points were amended and explained as presented below.

Sincerely

Prof. Richard D’Vries

Universidad Santiago de Cali

Reviewer #1:

In this review, the authors mainly summarized the progress in crystalline engineering aspects for f-elements- and ditopic-aliphatic-ligand based luminescent coordination polymers. Some important aspects, such as coordination polymers based on oxalate linker, flexible linkers with –Ch2-spacers, luminescent properties of f-element contained compounds were reviewed and discussed. Moreover, the perspective was also commented. After carefully reading full text, I feel that this manuscript is acceptable for publication after proper revisions. The complex is an isolated compound, so it is hard to think of it as a helical structure.

  1. In the abstract, the main points of this review should be included, and some general comments on the progress in this topic should also be added.

Response: The main points of this review were added to the final of the abstract section.

  1. A paragraph which presents the main content of this review should be added as the last paragraph of the INTRODUCTION section.

Response: The main points of this review were added to the final of the introduction section.

  1. The title of section 2 should be revised. Moreover, it is better to number the sub-sections in section 2.

Response: The section 2 was kept as a discussion but the subsections were renumbered as per the reviewer's suggestion

  1. Section 3 is significant. However, there are a few words for this section. The authors should read more published papers and add some new content for this section.

Response: The name of the section 3 was changed for “conclusions”. In this section as added a paragraph about the two new structures reported in the work.

  1. It is not satisfactory that only 59 references were cited. Some more references are asked for citations. Additionally, the references should be carefully formatted based on the journal criteria.

Response: The number of references was increased around 100 references.

  1. In general, English seems good to me. However, there are still some bad sentences and grammar errors. Therefore, the authors should carefully polish their English.

Response: Thank you very much for the comment, we correct grammatic errors and try to improve the English for recommendation of both reviwers. All the changes are highlighted in the text.

Reviewer 2 Report

The paper displays information about the preparation of lanthanide- and actinide-loaded coordination polymers with diverse degrees of crystallinity, architectures, dimensionalities… based on the choice of different organic ligands combined with luminescent 4f and/or 5f metal centres. The paper is well written and consistent/robust in terms of literature found by the Authors. The studies presented by the Authors are also quite interesting to the field, considering the information provided by them. The article presents some minor adjustments/revisions, which I’ve reported in the attached .pdf file. In conclusion, I recommend the publication of the paper in Molecules, after minor revisions.

Author Response

Dear  Felicia Yao

Assistant Editor

Molecules

Dear Felicia

In the following document I send to you the answers and modifications for the submitted Viewpoint to Molecules (1774737).

According to the strong suggestion of the referees, we put major efforts to considerably improve the manuscript to be accepted in Molecules. In this sense, we have carefully considered all comments and all raised points were amended and explained as presented below.

Sincerely

Prof. Richard D’Vries

Universidad Santiago de Cali

Reviewer # 2

The paper displays information about the preparation of lanthanide- and actinide-loaded coordination polymers with diverse degrees of crystallinity, architectures, dimensionalities… based on the choice of different organic ligands combined with luminescent 4f and/or 5f metal centres. The paper is well written and consistent/robust in terms of literature found by the Authors. The studies presented by the Authors are also quite interesting to the field, considering the information provided by them. The article presents some minor adjustments/revisions, which I’ve reported in the attached .pdf file. In conclusion, I recommend the publication of the paper in Molecules, after minor revisions.

  • Entire Paper:

â–ª Check commas and (unwanted) spaces through the paper. â–ª Change “4f” and “5f” to “4f” and “5f“ through the paper.

2) Title: â–ª Nothing to suggest to the Authors.

3) Abstract: â–ª Line 23 = change “of obtaining” to “to obtain”.

4) Discussion: â–ª Add numbering to sub-sections (i.e. 2.1, 2.2, 2.3…).

â–ª Check unwanted spaces in the chemical formula in the late sub-sections.

â–ª Line 33 = add “(CPs)” after “Coordination polymers”.,

â–ª Line 59 = add “Cambridge Crystallographic Data Centre” before “(CCDC)”.

â–ª Line 60 = add “increases” after “backbone”.

â–ª Line 61 = change “increments” to “increases”.

â–ª Line 69 = add a [REF] at the end of the phrase, if possible.

â–ª Line 70 = change “Ln-CPs or An-CPs” to “Ln-CPs or An-CPs”. Define Ln = lanthanides and An = actinides before. â–ª Line 73 = add a [REF] after “luminescent devices”, if possible. â–ª Line 78 = add “main” or “most intense” between “to the” and “for-“.

 â–ª Line 92 = add “luminescent” before “3D coordination”.

â–ª Line 94 = add “produced” after “Eu-MOF”.

â–ª Line 96 = remove “Recent”. â–ª Line 97 = What are the applications mentioned?

â–ª Line 105 = change the phrase to “components could open a wide research area where these compounds may be used as novel luminescent”.

â–ª Line 122 = add “to” after “possible”.

â–ª Add a [REF] at lines 124, 126 and 128, if possible.

â–ª Line 129 = add “they” before “generate”.

â–ª Line 134 = change “to forming” to “to the formation of”.

â–ª Line 164 = add a [REF], if possible. â–ª Line 170 = add “of the” before “crystalline”.

â–ª Line 176 = change “ 34 .” to “.[34]”.

â–ª Line 186 = add “region” after “near-infrared”.

â–ª Line 199 = add “, and” after “region”.

â–ª Line 243 = check if the reference “Figure 3” in the text is correct or not.

5) Materials and Methods:

â–ª Add numbering to sub-sections (i.e. 4.1, 4.2).

 â–ª Between Line 275 and 276 = a space between the end of the phrase and the start of the sub-section appears to be missing.

6) References:

â–ª Reference 47: check the reference because some information is missing (i.e. issue, volume, pages)

Response: All the suggestion were made and are highlighted in the manuscript.

Round 2

Reviewer 1 Report

All my concerns were well responded to, and the revisions sound satisfactory. Therefore, I recommend accepting it for publication in the present form.